# Health Information Source Patterns and Dietary Variety among Older Adults Living in Rural Japan

**DOI:** 10.3390/ijerph21070865

**Published:** 2024-07-01

**Authors:** Kumi Morishita-Suzuki, Shuichiro Watanabe

**Affiliations:** 1Sendai Center for Dementia Care Research and Training, Miyagi 989-3201, Japan; 2Graduate School of Gerontology, J. F. Oberlin University, Tokyo 194-0213, Japan

**Keywords:** health information sources, Japan aging populations, health behaviors, diet, latent class analysis

## Abstract

Dietary variety is associated with some health outcomes among older adults. Rural areas, however, often have difficulty accessing health information that influences dietary variety. This study aimed to identify patterns of health information sources by using latent class analysis and assess their association with dietary variety among older adults aged ≥ 75 in rural Japan (n = 411). Three patterns of health information sources were identified: multi-sources (29.7%), television-only (53.5%), and non-sources (16.8%). In the multi-sources pattern, more people used television, radio, and newspapers. The television-only pattern had mostly television users, with fewer other sources. The non-sources pattern had many reporting “none.” Logistic regression analysis revealed that the multi-sources pattern has a significant positive effect on dietary variety compared with the non-sources pattern (odds ratio: 5.434, 95% confidence interval: 1.792–16.472), even after adjusting for socioeconomic factors and physical health status. These findings underscore the positive impact of broad access to health information on the dietary habits of older individuals. The study highlights the importance of promoting access to diverse health information sources to enhance dietary variety and overall well-being among rural older adults.

## 1. Introduction

Dietary variety significantly affects health outcomes in older adults, including frailty [1,2,3], dementia [4], and mortality [2,5]. Dietary guidelines emphasize dietary variety as an essential indicator of nutritional quality because no single food provides the adequate nutrients required for optimal health [6]. Despite these benefits, aging is a risk factor for a lack of it. Specifically, older adults aged ≥ 75 years are more likely to have poor dietary variety [7,8]. Therefore, identifying modifiable factors and developing interventions to improve dietary variety in older adults is crucial.

Socioeconomic status (SES) is closely related to dietary variety in older adults. Higher educational attainment, income, and urban living are associated with better dietary variety [9,10,11,12,13,14,15,16]. Individuals with better SES often have greater access to health information, which influences their dietary behavior. A longitudinal study in China found higher perceived economic status positively affected dietary variety more in rural than urban areas [13]. Disparities in healthcare and healthy food access due to SES are more pronounced in rural and developing regions [5,15]. Therefore, efficient interventions should be considered to improve the eating habits of older adults living in rural areas.

Promoting health literacy is a critical and modifiable factor for improving dietary variety [17,18]. Health literacy is the ability to obtain, understand, and use health information and healthcare services to make reasonable health decisions [19,20]. Seeking health information is a foundational behavior in health literacy. With recent developments in information and communication technology (ICT), sources of health information have become more diverse. Older adults often rely on television, newspapers, families, and healthcare professionals for health information [21,22,23,24,25,26,27]. The sources of health information vary according to older adults’ demographics. For example, older adults living in urban areas are significantly more likely to use the Internet as a source of health information. Conversely, those living in rural areas tend to rely less on Internet sources [26]. Notably, the impact of different sources of health information on health literacy and health behaviors, including dietary variety, differs [23,24,25,27,28,29,30]. Mass media sources are not significantly associated with meeting diet recommendations, while community organizations positively influence them in older Americans [31]. In Thailand, healthcare professionals, the Internet, and newspapers significantly enhance health literacy, whereas community radio decreases it [24]. Therefore, understanding the sources of health information among older adults is valuable for designing interventions to improve dietary variety. 

Most studies examine health information sources individually, making it challenging to discern patterns of combined sources. As sources diversify, it is crucial to understand these patterns. For example, one can imagine patterns in which individuals exclusively prefer paper-based media, favor both paper and ICT, and rely on information from healthcare professionals, family members, and others. Investigating these patterns is essential for effective health information dissemination and improving dietary habits. To our knowledge, no study has identified these patterns and their association with dietary variety among older adults. 

This study aimed to identify patterns in health information sources and their association with dietary variety among older adults in rural Japan. We hypothesize the following: (1) Interpersonal sources such as family or neighbors are identified as primary health information sources because informal family support significantly influences health behaviors in rural areas [32]. (2) Older adults using diverse health information sources have better dietary variety than those using fewer sources. 

As of 2023, Japan is one of the most rapidly aging populations globally, with 29.0% of the population aged ≥ 65 years (15.5% of the population aged ≥ 75 years) [33]. With increasing urbanization, the population living in rural areas in Japan has decreased to ap-proximately 20% [34]. However, the rapid aging of the rural population highlights the pressing need for an efficient healthcare system [35]. Examining the association between health information source patterns and dietary variety among older adults aged ≥ 75 years living in a rural area may provide important suggestions for public nutritional approaches. These insights can help to augment the healthcare system.

## 2. Materials and Methods

### Study Design and Participants

This study used cross-sectional data from a prospective cohort survey, conducted in April 2019, of community-dwelling older adults aged ≥ 75 in Tsumagoi village, Gunma Prefecture, Japan. This cohort study is part of the Integrated Longitudinal Studies on Aging in Japan (ILSA-J), a nationwide cohort study of community-dwelling older adults in Japan [36]. They participated in an annual medical check-up held by the village. A questionnaire was mailed before the checkup and collected at the venue. Tsumagoi is located in western Gunma, a highland area. As of 2019, it had a population of 9521, with a population density of 28.2 people/km^2^. The proportion of people aged ≥ 65 was 36.4%, which is higher than the national average (28.4%) [37]. Agriculture is the main industry, with approximately 25% of the total population employed in agriculture [37].

## 3. Measures

### 3.1. Sources of Health Information

Multiple responses from the following options ascertained sources of health information: none, television, radio, books/magazines, newspapers, family living together, family living separately, friends, healthcare professionals, the Internet, and neighbors.

### 3.2. Dietary Variety Status

Dietary variety score (DVS) was used to measure dietary variety status [38]. The DVS was developed to assess the intake frequency of ten food categories (fish/shellfish, meat, eggs, milk and dairy products, soybean products, seaweed, potatoes, fruits, green/yellow vegetables, and fat/oil) that constitute a large proportion of the main and side dishes consumed daily in Japanese cuisine. Respondents rated their intake frequency for each food group using four options: (a) daily or almost daily, (b) once every two days, (c) once or twice a week, or (d) hardly ever. Option (a) was assigned one point, whereas options (b)–(d), which represent intermittent consumption, were assigned zero points. We calculated the DVS as the sum of the points so that the total score ranged from 0 to 10, with higher scores indicating greater variety in food intake. Individuals with a DVS of ≤3 tend to face a greater risk of muscle mass loss and diminished physical function. Those with a DVS of ≥7 exhibit a lower risk [39]. Therefore, a DVS ranging from 0 to 3 was categorized as indicating poor dietary variety, 4 to 6 as moderate, and ≥7 as high in this study.

### 3.3. Covariates

Covariates were age, gender (men or women), educational attainment (≤12 years or ≥13 years), economic status (not distressed or distressed), household (living alone or living with others), chronic disease (0 or ≥1), cooking habit (yes or no), chewing ability (robust or dysfunction), exercise habits (yes or no), accessibility to grocery stores (accessible or not accessible), and health literacy (yes or no). Cooking habits were assessed by asking, “Have you cooked meals in the past week?” Responses were dichotomized into no (“none”) and yes (“once a day”, “twice a day”, and “three times a day”). The physician checked chewing ability, and responses were dichotomized into two categories: “dysfunctional” if there were concerns about teeth, gums and bite, and “robust” if the patient could chew and eat anything. Shopping accessibility was surveyed using the following question: “Do you feel inconvenienced in daily grocery shopping?” Response options were either no (not accessible) or yes (accessible). Health literacy was evaluated by asking, “Can you judge the credibility of health-related information?” This was a question from the JST-Index of Competence questionnaire [40].

## 4. Statistical Analysis 

Latent class analysis (LCA) was used to identify patterns in health information sources. LCA enables researchers to evaluate the connection between observed data and hidden variables (latent classes) and distinguish them from multivariate categorical data. The latent classes identified by the LCA are categorical; thus, the cases in the sample can be categorized into comprehensive and distinct subsets [41]. We determined the number of classes using model fit statistics: a significant result for the Lo–Mendell–Rubin (LMR) test, a low Bayesian information criterion (BIC) and high entropy.

After determining the number of classes, we conducted univariate and multivariate analyses to examine the association between the classes and dietary variety. We used chi-square and Kruskal–Wallis tests for categorical and continuous variables, respectively. We also applied the Bonferroni correction when significant differences were observed in these analyses. After adjusting for covariates, a multinomial logistic regression analysis was conducted with the classes of health information sources as the independent variable and dietary variety status as the dependent variable. Dependent variables were a poor variety of food intake (DVS ≤ 3), a high variety of food intake (DVS ≥ 7), and the presence of “daily or almost daily” for each of the ten food items. Statistical significance was set at *p* < 0.05. SPSS version 29.0 (IBM Corporation, NY, USA) and Mplus version 8.8 (Muthen & Muthen, LA, USA) was used for all the statistical analyses.

## 5. Results

### 5.1. LCA of Health Information Sources

Table 1 presents the fit statistics for the model. We selected a three-class model for the following reasons: The three-class model exhibited the best BIC. Second, although the entropy was lower than ≥0.8, indicating high model fit, the three-class model had a significantly better fit than the two-class model, which had the highest entropy, as indicated by the LMR result.

Table 2 presents the percentage of each health information source by class. Participants were classified into Class 1 (29.7%), Class 2 (53.5%) and Class 3 (16.8%). Class 1 had high percentages of television, radio and newspapers. Considering that these high percentages across multiple sources are unique to Class 1, we labeled Class 1 as multi-sources. Class 2 comprised a high percentage of television users. We labeled this class television-only. Class 3 had a high percentage of “none”; therefore, we named it non-sources.

### 5.2. Univariate Analysis

Table 3 shows the differences in participants’ characteristics by class. Compared to individuals from non-sources, those from multi-sources and television-only were significantly more likely to be women and to have cooking habits. Additionally, individuals identified as having multi-sources were more likely to have higher health literacy than those in the other two classes.

Table 4 presents the differences in dietary variety by class. Significant differences were observed among the three classes for dietary variety and consumption of fish/shellfish, soybean products, potatoes, fruits, and green/yellow vegetables. Multiple comparisons showed that multi-sources had a significantly higher percentage of high dietary variety, eating fish/shellfish, and consuming green/yellow vegetables daily than non-sources. The multi-sources group also had significantly higher intake rates of soy products and potatoes than the other two classes. The non-sources group had a significantly higher percentage of poor dietary variety and a lower percentage of fruit intake than the other two classes.

### 5.3. Binary Regression Analysis

Table 5 summarizes the results of the binary logistic regression analysis. Participants who identified multi-sources and television-only exhibited significantly reduced odds of poor dietary variety and increased odds of high dietary variety compared to non-sources.

Table 6 presents the associations between health information source patterns and the intake of the ten food items. Across the ten food categories, individuals in the multi-sources group showed significantly elevated odds of consuming fish/shellfish, meat, soybean products, potatoes, fruits, green/yellow vegetables and fat/oil compared with those in non-sources. Similarly, participants in the television-only group demonstrated significantly increased odds of consuming fish/shellfish, meat, soybean products, fruits and fat/oil compared to non-sources.

## 6. Discussion

Notwithstanding the importance of health literacy, little is known about health information source patterns among older adults and their association with dietary variety. To our knowledge, this study is the first to identify the heterogeneity in patterns of health information sources and their relationship with dietary variety among older adults aged ≥ 75 in rural Japan. 

Studies have found that older adults living in rural areas are more likely to have poor health information and dietary variety [13,14,15,16,26,27]. In Japanese studies, 16.2% of older adults reported no particular health information source [27], and poor dietary variety (DVS ≤ 3) ranged from 23.5 to 55.4% in rural areas and 24.2–60.4% in urban areas [39,42,43,44,45,46]. The health information sources and dietary habits of our participants were consistent with these findings.

This study’s primary hypothesis, which was not supported, posited that interpersonal sources of health information emerge in a comparatively large proportion of older adults residing in rural areas. Our primary hypothesis that interpersonal sources of health information are predominant among rural older adults was not supported. The three patterns of health information sources were multi-sources, television-only, and non-sources. The percentages of interpersonal health information sources ranged from 5.9 to 22.9 %. According to the representative panel survey in Japan, the sources of health information among older adults aged ≥ 70 were the following: television (55.4%), family (30.9%), newspapers (28.5%), books/magazines (20.3%), friends (19.7%), healthcare professionals (16.8%), none in particular (16.2%), radio (6.3%) and the Internet (1.8%) [27]. Compared to this finding, it is evident that dependency on inter-personal sources was not substantial in our study population. Two plausible explanations for the minimal selection of interpersonal sources are as follows: First, interpersonal information sources are easily accessible but may lack reliability. The Internet provides more accessible information, but its use is significantly lower among older adults in this region. Conversely, books and printed materials are relatively accessible and may be preferred to interpersonal sources. Second, respondents habitually exchanged health information with family members and neighbors but may not have perceived them as sources of health information. Future studies should be qualitative in nature.

This study’s second main hypothesis, which was that individuals identified as having patterns with diverse health information sources were more likely to have a higher dietary variety, was supported. Our findings suggest that multi-sources or television-only patterns are more likely to produce a better dietary variety and less likely to produce a poor dietary variety compared to non-sources patterns. Additionally, the odds ratios for dietary variety in the multi-sources group were higher than those in the television-only group. Those accessing health information from books and newspapers had higher dietary variety, while mass media sources showed no significant association [30]. The results showed that print materials, which can be stored and referenced, may contribute to better dietary habits. 

Additionally, our findings showed that the patterns of health information sources were associated with the daily intake status of each of the ten food items. The multi-source group was more likely to consume fish, shellfish, meat, soybean products, potatoes, fruits, green/yellow vegetables, and fat/oil than the non-sources group. However, television-only had no significant association with the intake of fish/shellfish, potatoes or green/yellow vegetables. These three foods are rich in nutrients crucial for maintaining health in older adulthood. For instance, fish/shellfish and vegetable oils contain high levels of omega-3 fatty acids, which prevent cognitive and physical dysfunction [47,48]. Moreover, a deficiency in protein, which is abundant in fish/shellfish, and vitamins and minerals, which are found in green/yellow vegetables, can elevate the risk of sarcopenia [49]. Promoting diverse health information sources is essential for encouraging the intake of these nutritious foods.

Our study had two limitations: First, it used a cross-sectional design, and we could not infer causal relationships between the patterns of health information sources and dietary variety. Therefore, future studies should investigate how patterns of health information sources affect dietary variety among older adults using a longitudinal design. Second, participants were limited to a single rural area. Varying climates and industrial structures may yield distinct health information sources, dietary diversity and their associations. 

## 7. Conclusions

We discovered the effect of different patterns of health information sources on the dietary variety of older adults aged ≥ 75 in rural Japan. Contrary to our expectations, inter-personal sources such as family and neighbors were not primary options. We identified three distinct patterns: multi-sources, television-only, and non-sources. Individuals in the multi-sources group who accessed diverse health information showed a higher likelihood of having better dietary variety. This implies that access to a wider range of health information positively influences older adults’ dietary habits. These findings emphasize promoting access to various health information sources to enhance dietary variety and overall well-being among older adults living in rural areas.

## Figures and Tables

**Table 1 ijerph-21-00865-t001:** Model fit statistics of latent class analysis.

No. of Class	AIC	BIC	Entropy	Average Latent Class Probabilities		LMR Test
1	4266.713	4312.35	-	-		-
2	3837.581	3933.00	0.99	class1:	0.99	453.132 (*p* < 0.001)
				class2:	1.00	
3	3771.04	3916.23	0.76	class1:	0.82	90.544 (*p* < 0.001)
				class2:	0.89	
				class3:	0.99	
4	3743.802	3938.78	0.82	class1:	0.82	51.235 (*p* < 0.001)
				class2:	0.93	
				class3:	0.99	
				class4:	0.91	
5	3722.92	3967.68	0.80	class1:	0.79	44.882 (*p* < 0.001)
				class2:	0.91	
				class3:	0.83	
				class4:	1.00	
				class5:	0.87	
6	3724.483	4019.02	0.74	class1:	0.72	22.437 (*p* = 0.2200)
				class2:	0.81	
				class3:	0.89	
				class4:	0.71	
				class5:	0.98	
				class6:	0.75	

AIC: Akaike’s Information Criterion, BIC: Bayesian Information Criterion, LMR test: Lo–Mendell–Rubin test.

**Table 2 ijerph-21-00865-t002:** Health information sources by classes.

	Total	Class 1 Multi-Sources	Class 2 Television-Only	Class 3 Non-Sources
	(*n* = 411)	(*n* = 122)	(*n* = 220)	(*n* = 69)
None	17.3%	0.0%	0.9%	**100.0%**
Television	**75.9%**	**99.2%**	**84.5%**	7.2%
Radio	8.0%	27.0%	0.0%	0.0%
Books/magazines	27.7%	**89.3%**	2.3%	0.0%
Newspapers	33.1%	**61.5%**	27.7%	0.0%
Family living together	20.4%	18.0%	27.3%	2.9%
Family living separately	5.6%	8.2%	5.0%	2.9%
Friends	22.9%	36.9%	22.3%	0.0%
Healthcare professionals	10.5%	18.0%	9.5%	0.0%
Internet	2.7%	6.6%	1.4%	0.0%
Neighbors	9.7%	13.9%	10.5%	0.0%

Percentages of ≥50% are shown in bold to facilitate interpretation.

**Table 3 ijerph-21-00865-t003:** Differences in participants’ characteristics by class.

	Total (*n* = 411)	Class 1 Multi-Sources (*n* = 122)		Class 2 Television-Only (*n* = 220)		Class 3 Non-Sources (*n* = 69)		*p*
Gender								
Men	42.6%	31.1%	a	42.7%	a	62.3%	b	<0.001
Women	57.4%	68.9%		57.3%		37.7%		
Age								
Median (25%–75%)	79.0 (77.0–83.0)	79.0(76.0–82.0)		79.5 (77.0–84.0)		81.0 (77.0–84.0)		0.056
Education attainment								
≤12 years	65.7%	57.4%		68.2%		72.5%		0.056
≥13 years	34.3%	42.6%		31.8%		27.5%		
Economic statement								
Not distressed	91.5%	92.6%		92.7%		85.5%		0.149
Distressed	8.5%	7.4%		7.3%		14.5%		
Household								
Living with others	83.7%	86.9%		82.7%		81.2%		0.5
Living alone	16.3%	13.1%		17.3%		18.8%		
Chronic disease								
0	82.7%	82.8%		84.1%		78.3%		0.535
≥ 1	17.3%	17.2%		15.9%		21.7%		
Chewing ability								
Robust	71.5%	72.1%		69.5%		76.8%		0.498
Dysfunctional	28.5%	27.9%		30.5%		23.2%		
Cooking habits								
No	29.9%	19.7%	a	30.0%	a	47.8%	b	<0.001
Yes	70.1%	80.3%		70.0%		52.2%		
Exercise habits								
No	61.8%	57.4%		65.9%		56.5%		0.183
Yes	38.2%	42.6%		34.1%		43.5%		
Accessibility to grocery stores								
Accessible	62.0%	59.8%		62.3%		65.2%		0.759
Not Accessible	38.0%	40.2%		37.7%		34.8%		
Health literacy								
No	26.5%	16.4%	a	28.2%	b	39.1%	b	0.002
Yes	73.5%	83.6%		71.8%		60.9%		

Different letters, such as ‘a’ and ‘b’, indicate significant differences between the groups based on multiple comparisons. Groups with the same letter do not have significant differences.

**Table 4 ijerph-21-00865-t004:** Differences in dietary variety by classes.

	Total (*n* = 441)	Class1 Multi-Sources (*n* = 122)		Class2 Television-Only (*n* = 220)		Class3 Non-Sources (*n* = 69)		*p*
Dietary variety								
Poor (DVS ≤ 3)	46.5%	36.1%	a	44.5%	a	71.0%	b	<0.001
Medium (DVS 4–6)	35.8%	37.7%	ab	39.1%	a	21.7%	b	
High (DVS ≥ 7)	17.8%	36.2%	a	16.4%	ab	7.2%	b	
Percentage of “daily or almost daily”								
Fish/shellfish	34.8%	42.6%	a	34.1%	ab	23.2%	b	0.024
Meat	20.7%	21.3%		23.6%		10.1%		0.053
Egg	43.6%	45.9%		43.2%		40.6%		0.766
Milk/dairy products	56.9%	56.6%		57.3%		56.5%		0.989
Soybean products	51.1%	64.8%	a	50.0%	b	30.4%	c	<0.001
Seaweed	24.3%	29.5%		22.7%		20.3%		0.26
Potatoes	31.9%	44.3%	a	26.8%	b	26.1%	b	0.002
Fruits	49.4%	55.7%	a	52.7%	a	27.5%	b	<0.001
Green/yellow vegetables	52.1%	60.7%	a	51.8%	ab	37.7%	b	0.009
Fat/oil	27.0%	31.1%		28.2%		15.9%		0.064

DVS: Dietary Variety Score. Different letters, such as ‘a’ and ‘b’, indicate significant differences between the groups based on multiple comparisons.

**Table 5 ijerph-21-00865-t005:** The association between health information source patterns and dietary variety.

		Poor Dietary Variety ^a^			High Dietary Variety ^b^		
		OR	(95% CI)	*p*	OR	(95% CI)	*p*
Patterns of health information sources							
	Multi-sources (ref. Non-sources)	0.225	(0.111–0.458)	<0.001	5.434	(1.792–16.472)	0.003
	Television-only (ref. Non-sources)	0.292	(0.154–0.553)	<0.001	2.900	(1.007–8.347)	0.048
Gender	Women (ref. Men)	0.379	(0.215–0.666)	<0.001	3.566	(1.576–8.069)	0.002
Age	Years	0.929	(0.879–0.981)	0.008	1.075	(1.006–1.148)	0.032
Educational attainment	≥13 years (ref. ≤ 12 years)	0.779	(0.486–1.250)	0.301	1.409	(0.788–2.520)	0.247
Economic statement	Not distressed (ref. Distressed)	1.315	(0.603–2.866)	0.491	0.510	(0.144–1.808)	0.297
Household	Living alone (ref. Living with others)	1.156	(0.623–2.144)	0.647	0.706	(0.281–1.776)	0.459
Chronic disease	≥1 (ref. 0)	0.957	(0.543–1.687)	0.880	1.156	(0.569–2.348)	0.688
Chewing ability	Dysfunctional (ref. Robust)	0.892	(0.555–1.435)	0.639	1.549	(0.855–2.808)	0.149
Cooking habits	Yes (ref. No)	2.828	(1.487–5.379)	0.002	0.214	(0.091–0.507)	<0.001
Exercise habits	Yes (ref. No)	0.532	(0.340–0.833)	0.006	2.120	(1.219–3.686)	0.008
Accessibility to grocery stores	Not accessible (ref. Accessible)	0.769	(0.492–1.202)	0.249	0.770	(0.431–1.374)	0.376
Health literacy	Yes (ref. No)	0.463	(0.280–0.764)	0.003	1.814	(0.897–3.667)	0.097

OR: odds ratio, 95%CI: 95% confidence interval. ^a^: Poor dietary variety indicated, Dietary Variety Score (DVS) ≤ 3 (ref. DVS ≥ 4). ^b^: High dietary variety indicated, DVS ≥ 7 (ref. DVS ≤ 6).

**Table 6 ijerph-21-00865-t006:** The association between health information source patterns and intake of ten food items.

	Independent Variable: Multi-Sources (Ref. Non-Sources)			Independent Variable: Television-Only (Ref. Non-Sources)		
	OR	(95% CI)	*p*	OR	(95% CI)	*p*
Outcome: Intake status “daily or almost daily”						
Fish/shellfish	2.831	(1.358–5.903)	0.006	1.944	(0.994–3.799)	0.052
Meat	3.040	(1.125–8.220)	0.028	3.464	(1.381–8.69)	0.008
Egg	1.364	(0.712–2.615)	0.350	1.170	(0.655–2.089)	0.596
Milk/dairy products	0.844	(0.442–1.614)	0.609	0.974	(0.548–1.732)	0.929
Soybean products	4.079	(2.034–8.179)	<0.001	2.357	(1.268–4.381)	0.007
Seaweed	1.752	(0.812–3.780)	0.153	1.208	(0.597–2.445)	0.599
Potatoes	2.834	(1.372–5.853)	0.005	1.122	(0.578–2.179)	0.734
Fruits	2.617	(1.302–5.259)	0.007	2.775	(1.471–5.236)	0.002
Green/yellow vegetables	2.176	(1.128–4.195)	0.020	1.723	(0.96–3.093)	0.069
Fat/oil	2.251	(1.005–5.041)	0.049	2.107	(1.002–4.431)	0.049

OR: odds ratio, 95% CI: 95% confidence interval. Adjusted for gender, age, educational attainment, economic statement, household, chronic disease, cooking habits, chewing ability, exercise habits, accessibility to grocery stores and health literacy.

## Data Availability

The datasets presented in this article are not readily available because this would compromise participant confidentiality.

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
