# Peer review of "Health Information Source Patterns and Dietary Variety among Older Adults Living in Rural Japan"

_ijerph, 2024, doi:10.3390/ijerph21070865_

Round 1

Reviewer 1 Report

Comments and Suggestions for Authors

This study used latent category analysis to identify patterns of health information sources and evaluate their association with dietary diversity among elderly people aged 75 and above in rural Japan. The article indicates that obtaining broader health information will have a positive impact on the dietary habits of elderly people, emphasizing the importance of promoting access to diverse sources of health information to enhance the dietary diversity and overall happiness of rural elderly people. The article contributes to promoting the health of elderly people in rural areas, but there are still the following issues in the article:

1. The research sample of the article is limited to one rural area in Japan (Tsumagoi Village, Gunma Prefecture). Can the data obtained represent the entire Japan. Due to being located in a highland, will the obtained data be affected by objective factors such as climate?

2. The article's wording is too redundant and contains a lot of duplicate data.

3. The article categorizes health information sources into three categories: multiple sources, television only, and no sources. Is this not rigorous enough?

The explanation in the article discussion that the proportion of interpersonal sources of health information for rural elderly people is relatively small lacks persuasiveness.

5. The summary of the content in the article is not clear enough and lacks information coverage.

Reviewer 2 Report

Comments and Suggestions for Authors

Dear Authors,

Thank you for providing this evidence on unhealthy eating behaviors. This study sheds important light on the connection between older individuals' access to health information and their varied diets in rural Japan. The study underscores the necessity of improving rural residents' access to a wide range of health information to enhance the nutrition and well-being of older adults. Additionally, I have minor comments which can be found below:

ABSTRACT

Line 13: Authors mention some variables were “significantly associated with higher dietary variety”. Moreover, being this a regression, it should be said that those variables have a significant effect over the dependent variable.

INTRODUCTION

The 3rd paragraph is too expense. It could be divided into two paragraphs making it more smoot. On line 52, after citation 26 the paragraph can be finished and start the new one with the next sentence.

MATERIALS AND METHODS

Was the prospective cohort survey of community-dwelling older adults previously validated? If so, validity and reliability need to be included. If the survey was not validated, it also needs to be mentioned as a limitation or potential limitation of the study.

MEASURES, STATS, RESULTS, CONCLUSION

In general, this section is well structured and described.

DISCUSSION

Line 212-213: When mentioning the study is the fist in identifying the heterogeneity in patterns of health information sources in the specific population in Japan, the authors need to cite similar studies in different populations and or setting. Alternatively, authors can state something like “this is the first study to our knowledge”

The way each hypothesis, and then the general main findings was discussed by authors is well structured and helps the reader to keep the line of the research.

Round 2

Reviewer 1 Report

Comments and Suggestions for Authors

The author answered all my questions. Reasonable explanations have been provided for some of the issues that have arisen in the research. The quality of the revised manuscript has greatly improved.